# Selected Properties of Veneered Lightweight Particleboards with Expanded Polystyrene

**DOI:** 10.3390/ma15186474

**Published:** 2022-09-18

**Authors:** Pavlo Bekhta, Ruslan Kozak, Ján Sedliačik, Vladimír Gryc, Václav Sebera, Liubov Bajzová, Ján Iždinský

**Affiliations:** 1Department of Wood-Based Composites, Cellulose, and Paper, Ukrainian National Forestry University, 79057 Lviv, Ukraine; 2Department of Wood Science and Technology, Faculty of Forestry and Wood Technology, Mendel University in Brno, Zemědělská 3, 613 00 Brno, Czech Republic; 3Department of Furniture and Wood Products, Technical University in Zvolen, T.G. Masaryka 24, 960 01 Zvolen, Slovakia; 4Department of Wood Technology, Technical University in Zvolen, T.G. Masaryka 24, 960 01 Zvolen, Slovakia

**Keywords:** lightweight particleboards, expanded polystyrene, veneer, density, press temperature, properties

## Abstract

The aim of this study was to improve the properties of lightweight particleboards by their veneering. The industrially produced wood particles, rotary-cut birch veneer, expanded polystyrene (EPS) granules and urea-formaldehyde (UF) resin were used to manufacture non-veneered and veneered boards in laboratory conditions. The boards were manufactured with different densities of 350, 450 and 550 kg/m^3^ and with various levels of EPS content 4, 7 and 10%. Boards without EPS granules as the reference were also manufactured. Bending strength (MOR), modulus of elasticity in bending (MOE), internal bond (IB) strength, thickness swelling (TS) and water absorption (WA) of lightweight particleboards were determined. This study confirmed that veneering of lightweight particleboards by birch veneer improved mechanical properties significantly. The MOR and MOE of veneered boards throughout the whole density range of 350–550 kg/m^3^ meet the requirements of the CEN/TS 16368 for lightweight particleboards types LP1 and LP2. The IB strength of veneered boards only with density of 550 kg/m^3^ meets the requirements of CEN/TS 16368 (type LP1). The MOR, MOE and IB of non-veneered boards also meet the requirements of CEN/TS 16368 (type LP1) except boards with density of 350 kg/m^3^ for MOR and MOE, and except densities of 350 and 450 kg/m^3^ for IB.

## 1. Introduction

One of the acute problems faced by the manufacturers of particleboards during the last decade is the insufficient supply of wood raw materials. This problem is due to the overexploitation of forests and, accordingly, the growing demand for wood from various branches of the woodworking and building industry, as well as the energy sector, which uses wood as biomass, and increased price of wood material. One of the ways to solve this problem is to reduce the use of wood in the production of wood products by a production of wood-based materials with a reduced density, compared to the density of standard boards. It is well-known that wood-based materials with high density have some disadvantages: rapid tool wear, material and transportation costs, handwork and high weight of construction [1]. Therefore, such an approach to find the solution of the specified problem is important from both an ecological and economical perspective. From the ecological point of view, this approach allows saving wood, preserving the forest environment and reducing the pressure on the world’s remaining natural forests. From the economical point of view, the lightweight boards have several advantages for manufacturers, designers and consumers, including flexibility in design (use of thick elements, easier installation), easier transportation and handling and both lower transportation and raw material costs [2,3]. A research study showed that weight was ranked third (after design and price) in the priority list of customer attributes for buying household furniture [4].

However, it should be remembered that the manufacture of lightweight particleboards by simply reducing density results in decreased board properties in addition to other challenges such as surface finishing and post-forming difficulties. The operational properties and, above all, the mechanical characteristics of wood-based materials depend significantly on their density. The lower the density, the lower the durability and stiffness of the material; the proportion of small voids and pores increases significantly, which complicates their processing. The average density of conventional particleboards usually ranges between 600 kg/m^3^ and 750 kg/m^3^ [5]. According to CEN/TS 16368 [6], the light particleboards include boards whose density is less than 600 kg/m^3^.

There are various trends in reducing the density of wooden boards—the use of different wood species, in particular, low-density wood species such as poplar and paulownia [7,8], the use of annual or perennial plants [9,10], a combination of wood raw materials and agricultural fibers [10,11], the use of foamed adhesives [12], production of extrusion boards [13,14], making sandwich structures [15] and a combination of different concepts [16]. The extrusion method is one of the oldest technologies for weight reduction in particleboards [13]. A lightweight tubular fiberboard with a density of 550 kg/m^3^ weighs approximately 30 percent less than conventional MDF [14]. Sandwich panel construction may generate properties comparable to conventional panels with a significant density reduction [15]. Lyutyy et al. [1] found that the use of EPS enables the manufacture of lightweight wood-polymer composites (WPC) throughout the density range of 500–700 kg/m^3^, which is almost twofold less than the density of the conventional WPC.

Incorporation of expanded polystyrene (EPS) granules in the core layer of particleboards to replace part of the wooden particles is another strategy to achieve lightweight boards [3,10,16,17]. EPS granule is a material with a very low density (15–25 kg/m^3^) that contains 98% air and only the rest is polystyrene [18]. Doroudiani et al. [19] argue for the low consumption of wood in favor of polystyrene. Dziurka et al. [16] concluded that substituting 7% of wood particles with EPS granules in the core layer makes it possible to manufacture lightweight particleboards with a density from 500 kg/m^3^ to 650 kg/m^3^. On the other hand, Meinlschmidt et al. [10] found that using up to 8% of expandable filler did not have a positive effect on the properties of lightweight particleboards with a density below 500 kg/m^3^. Whereas, other authors [3] showed that the properties of the lightweight boards do not undergo significant changes even with an increase in the amount of expandable filler to 15%. The literature shows that lightweight wood-based boards using expanded polystyrene may be produced from different amounts of EPS added to wood particles [3,10,16,20] and from different EPS granule diameters [21]. EPS granules are also added for the production of various types of particleboards [3,10,16,20], fiberboards [21] or even lightweight wood plastic composites [1].

Different authors propose alternative pathways to improve the properties of lightweight wood-based boards, namely by adjusting particle geometry (shape and dimension) in the core layer [22], core layer particle orientation [23], asymmetrical distribution of densities [24] or optimization of processing conditions [15,25,26]. Usually, overlaying of wood-based boards with wood veneer sheets improves their appearance and properties resulting in value-added products [27]. These authors found that furniture panels veneering with one layer of sliced wood veneer increases its bending strength (MOR) by 63%. Moreover, the veneering of particleboard surfaces with wood veneer was effective for reducing the formaldehyde emissions of the boards [28]. For the improvement properties of lightweight boards, different materials could also overlay them [29]. The veneered lightweight particleboards had better or similar MOR and modulus of elasticity (MOE) compared with commercial products [30,31]. The lightweight veneered fiberboards with densities of 300–500 kg/m^3^ had good dimensional stability and two and four times higher MOR and MOE, as those of homogeneous fiberboards, respectively [32]. Previous authors had also shown [33,34] that veneering of WPC with rotary-cut birch and sliced oak veneers improved the MOR in 1.9–2.6 and 7.9–10 times across and along veneer fibers, respectively, for both types of veneer compared with non-veneered WPC. The highest values of MOR were observed in WPC veneered with rotary-cut birch veneer.

Nowadays, there is very little research on the veneering of lightweight boards using EPS granules. Therefore, this study aimed to improve the properties of lightweight particleboards with densities ranging from 350 to 550 kg/m^3^ using EPS granules for the core and birch veneers for the faces. Results were compared with relevant standards for lightweight particleboards and previous research.

## 2. Materials and Methods

### 2.1. Materials

The experiments used industrially produced rotary-cut birch (*Betula verrucosa* Ehrh.) veneer (LLC «ODEK» Ukraine, Orzhiv, Ukraine) with a thickness of 1.5 mm and wood particles (KRONOSPAN, s.r.o., Zvolen, Slovakia) that consisted of deciduous (20%) and coniferous (80%) wood species (Figure 1). The moisture content of birch veneer and wood particles, determined by the drying-weighing method, were 6 ± 2% and 2 ± 1%, respectively. Expanded polystyrene (EPS) granules with diameters of 4–8 mm and glass transition temperature of around 100 °C were purchased from the local company (honest.company s.r.o., Stara Tura, Slovakia).

The commercial urea-formaldehyde (UF) resin Dukol (Ostrava s.r.o., Czech Republic), hardener ammonium sulfate ((NH_4_)_2_SO_4_) and paraffin emulsion were used to prepare UF adhesive for the blending process with wood particles. The following UF adhesive recipe was prepared: UF resin—100 wt%, hardener—5 wt% and paraffin emulsion—0.85 wt%. For veneering the boards, the following UF adhesive recipe was prepared: UF resin—100 wt%, hardener—4 wt% and wheat flour (as a filler)—11.5 wt%. A 20% aqueous solution of hardener was added to both UF adhesive systems.

### 2.2. Manufacturing of Lightweight Particleboards

The lightweight single-layer particleboards of 300 × 300 mm dimensions and a thickness of 18 mm with a target density of 350, 450 and 550 kg/m^3^ were produced (Figure 2). The amount of UF adhesive was 10 wt% based on the total mass of oven-dried wood particles and EPS. The amounts of EPS were 0, 4, 7 and 10% based on the oven-dried mass of wood particles. The blending of UF adhesive, wood particles and EPS granules was carried out in a laboratory drum mixer. First, the wood particles and one half of UF adhesive were loaded into the drum and mixed for 10 min. After that, the EPS granules and the rest of UF adhesive were added and then this mixture was continually blended during the additional 5 min to obtain the homogeneous composition.

To produce the non-veneered lightweight single-layer particleboards (Figure 2a), the resinated particles were hand spread evenly into a 300 mm × 300 mm wooden box with a caul plate as the base to form the mat. To produce the veneered lightweight single-layer particleboards (Figure 2b), the bottom sheet of veneer with an applied adhesive layer was placed on a caul plate in a wooden box. Then, the resinated mixture of particles and EPS granules were poured on the veneer side with adhesive layer, and finally, the upper sheet of veneer with an applied adhesive layer was placed on the top of resinated mixture. The formed mat was then pre-pressed manually in the wooden box and after that in the cold press during 10 min to consolidate the thickness (Figure 3a). Next, the mat was subjected to hot pressing in an automatically controlled hydraulic laboratory press CBJ 100-11 (TOS, Rakovník, Czech Republic) (Figure 3b). The heating plate temperature was 200 °C, unit pressure 2.4 MPa and pressing time 14 s per mm of board thickness. This pressing temperature is mainly used in practice to produce the conventional particleboards, as well as by some authors [3] to manufacture the lightweight boards. The experimental design for this study is summarized in Table 1. Boards without EPS granules as the controls were also manufactured at the same pressing conditions.

### 2.3. Particleboards Testing

Three boards were produced for each type of particleboard in the experimental design. The reference particleboard was made without EPS. The boards conditioned at a relative humidity of 65 ± 5% and 20 ± 2 °C were cut into the samples with required testing size according to relevant standards and the following parameters were assessed:–Density according to EN 323 [35];–MOR and MOE according to EN 310 [36];–tensile strength perpendicular to the plane of the board—“internal bond” (IB) strength according to EN 319 [37];–thickness swelling (TS) after 24 h according to EN 317 [38] and water absorption (WA).

### 2.4. Measuring the Duration of Mat Heating

One of the key pressing parameters of the particleboards is the duration of mat heating to the core layer temperature necessary for the hardening of adhesive. To measure the duration of heating, a thermocouple TR-01A (Scientific and Manufacturing Association “Thermoprylad”, Lviv, Ukraine) was placed in the core layer of the mat and was connected to a digital multimeter UT33C (Scientific and Manufacturing Association “Thermoprylad”, Lviv, Ukraine). Data were collected when the surface mat contacted the pressure. The core temperature was recorded at every second until it reached 105 °C and all data were saved to a computer.

### 2.5. Statistical Analysis

The results were analyzed using STATISTICA 12.0 package. The performed analysis was based on analysis of variance (ANOVA) and homogeneous groups were distinguished with the use of Duncan’s Range tests. The results were analyzed on the significance level of *p* = 0.05.

## 3. Results

### 3.1. Heating of the Mat

The temperature inside the core layer of mat during hot pressing is important for the chemical and physical processes that contribute to the melting and penetrating EPS granules and bonding between resinated wood particles. It was found that the mat with EPS content of 7% warms up the fastest, and the slowest one without EPS (Table 2). The value of the duration of heating of the core layer of the mat up to 100 °C is 12–33% less for the boards with 7% EPS content than for the boards without EPS. At 4% and 10% content of EPS in the mat, the rate of its heating differs slightly. It is obvious that the rate of heating depends on a number of factors, in particular, on the number of contacts between the wood particles and EPS granules. There is a significant proportion of voids in the mat with 0% content of EPS that do not conduct heat well. That follows from the microscopic observations of the microstructure of lightweight particleboards using EPS [20]. These authors showed that the incorporation of EPS granules filled in the voids between resinated wood particles improved the core layer integrity. This generates a more favorable internal structure through which heat is transferred from the surface to the core layer of the mat. However, an increase in EPS content of more than 7% causes a slowdown of mat heating. This is because at 10% EPS content, the volume of granules is greater than the volume of particles, and the thermal conductivity of EPS is lower compared to the wood particles. Therefore, in order to reduce the pressing time of lightweight particleboards, it is advisable to have not too high of an EPS content. In this study, this is 7% content of EPS, especially since such fillers are relatively expensive.

As expected, when the density of lightweight particleboards increases from 350 kg/m^3^ to 550 kg/m^3^, the duration of mat heating is shortened (Table 2). This is because the larger proportion of wood particles per unit volume of the mat has lower porosity and, as a result, increased thermal conductivity, which accelerates its heating. However, with a high density of the boards, their weight increases, which is contrary to the tendency to reduce the density of boards.

### 3.2. The Effect of Density and EPS Content on the Properties of Boards

ANOVA analysis showed that board density and EPS content significantly affect the mechanical (MOR, MOE, IB—except EPS content) and physical (TS and WA) properties of lightweight veneered particleboards. In addition, the effect of density was stronger than effect of EPS content. Similar results regarding the effect of EPS granules on the physical and mechanical properties of low-density boards were observed by other authors [3].

The following trends were observed for MOR—(1) a decrease in the density of samples leads to the decrease in the MOR; (2) increasing the EPS content in the board from 0% to 7% decreases the MOR. It was found that MOR of the samples for all EPS contents decreases linearly with a decrease in their density (Figure 4a). With decreasing density of samples from 550 to 350 kg/m^3^ and from 550 to 450 kg/m^3^ the average values of MOR decreases by 2.64 and 1.54 times, respectively. The addition of EPS to the wood particles in the amount of 4% and 7% makes it possible to manufacture boards with MOR values comparable to the MOR values of reference boards (without EPS content). Boards with 7% of EPS granules have slightly lower MOR values than boards with 4% of EPS granules, but the difference between these values is not significant (*p* > 0.05) based on Duncan’s test results. The addition of 4% or 7% EPS granules leads to a drop in MOR by 1.3 times. Whereas, the addition of 10% of EPS granules contributes to the dramatic drop in MOR by 3.7 times. Other authors [10] also observed a similar negative effect of using up to 8% EPS on the properties of lightweight particleboards with a density of 500 kg/m^3^. The contrary results obtained in the study [3] showed the use of EPS granulates positively influenced MOR and MOE. The authors explained this by using an initial densification during the first phase of hot pressing. Luo et al. [20] also found that adding EPS bead significantly increased the MOR and MOE compared with the control, which was related with the improved density profile; however, the variation in the number of EPS from 2.5% to 12.5% had no significant effect on the MOR and MOE.

The effect of the number of EPS granules on MOR can be explained as follows. Replacing the wood particles with 4% of EPS granules in the lightweight boards with significant inter-particle porosity does not change the volume of the mat, but increases its specific surface and reduces the proportion of the resinated particles. In the process of forming such a mat, the EPS granules are placed in the voids between the particles and during the process of pressing the mat; they have little effect on its compaction. The number of inter-particle adhesive contacts decreases and, as a result, the MOR of the boards decreases. During the pressing of boards, the EPS granules contribute to the compaction of particles and the increase in inter-particles adhesive contacts. As a result, the MOR of boards increases compared to the values of such MOR in the boards with 4% EPS content and reaches the values of the MOR of boards with 0% of EPS.

The influence of the density and EPS content on the MOE (Figure 4b) of the boards is similar to MOR. With decreasing density of samples from 550 to 350 kg/m^3^ and from 550 to 450 kg/m^3^, the average values of MOE decreases by 1.64 and 1.21 times, respectively. The lowest value of MOE was recorded for boards with 10% EPS granules (1759.9 MPa), which is 1.9 times less than for reference boards (3338.8 MPa). MOE values of boards with 4% and 7% content of EPS granules do not differ significantly (*p* > 0.05) based on the results of the Duncan test. With a lower density of boards and a higher content of EPS granules, the lightweight veneered boards are more elastic (flexible). On the contrary, several authors [16,39] showed that the addition of EPS granules significantly enhanced the MOE of the boards. These authors explained this by improved core layer integration and increased the rigidity of the core layer of the board, which can promote the stress transfer between the wood particles and accordingly enhances the MOE. Whereas, it was mentioned earlier that increasing the number of EPS beads from 5 to 15% revealed no significant effect on the MOE and MOR [3].

However, despite the decrease in the MOR values of veneered lightweight boards with EPS, the values of the MOR, as well as the MOE, are higher than the normative values of these parameters for lightweight particleboards according to the standard CEN/TS 16368 [6] in the entire investigated range of board densities and EPS content (Table 3). Moreover, the minimum requirement of MOR and MOE for conventional particleboards according to EN 312 (type P2) [40] (MOR > 11.5 MPa, for MOE is not regulated) and EN 312 (type P3) [40] (MOR > 13.0 MPa and MOE > 1600 MPa) is fulfilled for reference boards and lightweight boards with EPS for densities 450 and 550 kg/m^3^ (Figure 4).

Examinations of the tested board samples revealed that in all cases, the failures occurred in the axial plane in the core layer of the samples through delamination under the action of shear stress (Figure 5). This type of failure is explained by the presence of a strong peeling veneer in the outer layers of the board. During the bending test, no bonding failure occurred between the veneer and the core layer.

ANOVA analysis showed that board density and EPS content effect significantly the IB strength of lightweight veneered particleboards. IB values for the boards are presented in Figure 6. With decreasing density of samples from 550 to 350 kg/m^3^ and from 550 to 450 kg/m^3^, the average values of IB decreases by 2.53 (153%) and 1.52 (51.6%) times, respectively. This could be caused by an increased amount of low compacted particles in the core layer. A significant increase in IB values was also observed in the work [3] between the board densities 450 and 500 kg/m^3^. The lowest average value of IB 0.20 MPa was recorded for boards with 7% content of EPS granules, and the highest average values 0.25 Mpa and 0.22 Mpa for the boards with 4% and 10% content of EPS granules. The higher values of IB strength of the samples with 4% and 10% content of EPS granules compared to the values of IB for reference samples (without EPS) can be explained by analyzing micrographs of the microstructure of the boards [20,39]. As can be seen from the micrographs, the voids between the wood particles in lightweight boards are the source of breaks in the structure of the core layer. According to these authors, such breaks act as defects and contribute to the failure of bonds between the wood particles. Therefore, the filling of empty spaces between the wood particles with EPS granules makes the structure of the core layer more homogeneous and continuous. This improves consolidation and bonding in the core, which definitely improves the IB strength of the board. Shalbafan et al. [3] reported that the lightweight particleboards made using EPS fillers have nearly two times higher IB values than reference boards. Nevertheless, it should be noted that the average IB values for boards with 4, 7 and 10% content of EPS granules do not differ significantly among themselves, as well as between them and reference boards. In other studies, the adding of EPS beads ranging from 2.5% to 12.5% [20] or up to 15% [39] was shown to significantly increase the IB of the boards compared to the reference.

It is well-known that wood is polar, and polystyrene is a non-polar substance [41]. In this regard, the formation of chemical bonds between wood and EPS granules is impossible. Therefore, since wood is a porous material, the most likely bonding mechanism is mechanical bonding or interlocking of EPS granules between the wood particles. This follows both from this study and from the studies of other authors [3,10,11,15,16,20,39].

TS and WA values for 24-h immersion in water are illustrated in Figure 7. After 24 h of exposure in water, the samples with a higher density swell more. With decreasing density of samples from 550 to 350 kg/m^3^ and from 550 to 450 kg/m^3^, the average values of TS for 24 h decrease by 1.35 (34.7%) and 1.11 (10.7%) times, respectively. This can be explained by an increased density of the boards due to the increased proportion of swellable wood particles, by the long-term effect of water, which destroys the adhesive bonds, weakens the internal structure, increases the number of restorative deformations in the particles and, as a result, increases swelling of the boards. However, this is contradicted by observations of Shalbafan et al. [3] who observed a lower TS value due to less voids existing in the heavier boards.

The lightweight particleboards with EPS granules had lower TS values as compared with the reference boards. The lowest average value of TS 11.73% was recorded for boards with 10% content of EPS granules, and the highest average value 12.95 MPa for the reference boards without EPS granules. Nevertheless, it should be noted that the TS values for boards with 4%, 7% and 10% content of EPS granules do not differ significantly among themselves, but differ significantly with reference boards. Other authors [20] also observed no significant difference between TS of boards with different contents (from 2.5% to 12.5%) of EPS. The reduction in TS of the boards containing EPS is explained by the greater plasticity of EPS compared to wood particles and, as a result, the reduction in internal stresses in the boards with EPS after pressing. In addition, EPS as an inherent hydrophobic material, unlike wood particles as hydrophilic material with many polar hydroxyl groups (OH), does not swell in water.

Soaking the samples in water for 24 h causes water saturation by the wood particles, weakening the adhesive joints, swelling the boards and, as a result, increasing water permeability inside the structure of the board. Reference boards had significantly higher WA values than lightweight boards with EPS granules. WA values of the samples after 24-h immersion in water decreases with increasing density and EPS content (Figure 7b). With decreasing density of samples from 550 to 350 kg/m^3^ and from 550 to 450 kg/m^3^, the average values of WA for 24 h increase by 1.68 and 1.23 times, respectively. Such an increase in WA occurs due to an increase in the porosity and permeability of the boards with a decrease in their density. Several authors [26] also found that with increasing board density, the dimensional stability is significantly decreased because of the increased share of wood particles per unit volume of the board. With increasing content of EPS granules from 0% to 10%, the average values of WA for 24 h decrease by 1.48 times. This is explained by the hydrophobicity of EPS that does not shrink or swell when in contact with water [42]. Therefore, the water cannot penetrate after pressing into the EPS granules, but can only penetrate through the small voids between the fused EPS granules [26]. The granule fusion during the pressing process determines the water resistance of the EPS [42]. The greater the number of EPS granules that replace the wood particles in the board, the greater the reduction in WA. The lowest average values of WA 62.70% and 69.35% were recorded for boards with 10% and 7% content of EPS granules, respectively, and these values do not differ significantly. The highest average value of WA 93.0% was for the reference boards without EPS granules. The WA values for boards with EPS granules differ significantly with the WA values for reference boards without EPS granules. Thus, the bonding strength of the boards mainly depended on the level of mechanical locking of EPS to the wood particles, whereas WA and TS were mostly affected by the EPS distribution [16,39].

### 3.3. The Effect of Pressing Temperature on the Properties of Lightweight Particleboards

The veneered boards were manufactured with EPS content of 7% at two pressing temperatures, 200 °C and 220 °C. It was found that an increase in the pressing temperature from 200 °C to 220 °C negatively affects the MOR (Figure 8a). This indicates that high temperature leads to brittleness and lower resistance to static bending of the peeled veneer. It is known that the surface layer provides the MOR of the boards. In addition, with the increase in the density of the boards, the difference between the average values of MOR decreased. In particular, for the density of 350 kg/m^3^ and pressing temperature of 200 °C and 220 °C, the average values of MOR were 3.3 MPa and 1.2 MPa, respectively. Whereas, at the same pressing temperatures, but a density of 550 kg/m^3^, the average values of MOR were 28.8 MPa and 19.1 MPa, respectively. Thus, for the target densities of 350 and 550 kg/m^3^ this difference was 63.6% and 33.7%, respectively.

Figure 8b illustrates the influence of density and pressing temperature on the IB strength of the boards. It was found that the IB strength increases with an increase in the pressing temperature, except for the density of 450 kg/m^3^. Thus, with an increase in the pressing temperature from 200 °C to 220 °C, the values of IB strength increase for the density of boards 350 kg/m^3^ and 550 kg/m^3^ by 16.8% and 12.5%, respectively. It can be assumed that at a higher pressing temperature, EPS granules melt better and penetrate better into the cavities of the mat, resulting in small voids between the particles and improving the contact area, hence the bonding area among the particles and, therefore, providing higher IB strength. Moreover, it was found that a higher pressing temperature can also decrease the proportion of hydrophilic groups (OH) in wood, thus enhancing the interfacial compatibility between hydrophilic wood and hydrophobic polymer granules [43]. Shalbafan et al. [26] also observed that by increasing the pressing temperature from 130 to 160 °C and above, a better fusion of EPS cells with a smaller void volume between the cells can be achieved. Opposite results are obtained by authors [20] who claim that lower temperatures are better to avoid softening and shrinkage of the EPS granules. For boards with high EPS contents (10% and 12.5%), the high temperature (140 °C) had a negative effect on the properties of the boards.

Therefore, taking into account the decrease in the MOR with a slight improvement in the IB strength, as well as the increase in energy consumption with an increase in the pressing temperature, it is not advisable to increase the pressing temperature of lightweight veneered particleboards to more than 200 °C.

### 3.4. The Effect of Veneering on the Properties of Boards

Veneering of lightweight particleboards not only improves their aesthetic appearance, but also changes their structure and physical and mechanical properties. In this study, non-veneered (Figure 2a) and veneered (Figure 2b) lightweight particleboards with the number of EPS 7% were manufactured under conditions described in Section 2.2.

As it could have been expected, the veneering of boards improved MOR and MOE significantly (Figure 9). It was found, that the values of MOR were in 3.6, 4.7 and 5.2 times, and MOE were in 5.6, 5.4 and 5.2 times higher for veneered boards than for non-veneered boards with densities of 350, 450 and 550 kg/m^3^, respectively. Moreover, the values of MOR and MOE increased with rising density of non-veneered and veneered boards. As the density increased from 350 kg/m^3^ to 550 kg/m^3^, the values of MOR increased by 2.5 and 3.6 times and MOE increased by 2.1 and 1.9 times for non-veneered and veneered boards, respectively. Kawai et al. [30] also found that MOR of veneer-overlaid particleboards of densities 300–400 kg/m^3^ was much improved. In another work [32], the MOR and MOE of veneer-overlaid low-density (400–500 kg/m^3^) fiberboards were two and four times as much as those of homogeneous fiberboards, respectively. Other authors [11] observed a four-fold increase in MOR of lightweight veneered particleboards from wood and rape straw compared to the veneered conventional particleboards. It is well-known that MOR is proportionally correlated with boards density [44].

It was found that values of MOR and MOE of non-veneered (except density of 350 kg/m^3^) and veneered (throughout the density range) lightweight particleboards with 7% content of EPS meet the requirements for lightweight particleboard type LP1, according to CEN/TS 16368 [6] (MOR > 3.5 MPa, MOE > 500 MPa) (Figure 9). Moreover, according to this standard [6], the lightweight veneered boards could be also classified as type LP2 (MOR > 7.0 MPa, MOE > 950 MPa).

As expected, veneering did not affect IB strength of lightweight particleboards. It was found that IB rises proportionally as density of boards increases (Figure 10). The obtained results are in good agreement with the well-known statements that the density not only affects, but also determines the IB strength of wood-based boards [44]. A comparison of the IB values of lightweight boards with the requirements of standard CEN/TS 16368 [6] showed that only non-veneered boards with a density of 550 kg/m^3^, and veneered boards with densities of 450 and 550 kg/m^3^ meet the requirements of this standard for LP1 type boards. Whereas, the IB values of non-veneered boards with densities of 350 and 450 kg/m^3^ and veneered boards with a density of 350 kg/m^3^ did not meet the requirements of this standard. The explanation may be that expanded polystyrene is the least strong material among the components of boards, and its proportion in the relation to the proportion of wood particles is higher in the boards with densities of 350 or 450 kg/m^3^ than in the boards with a density of 550 kg/m^3^. This was also confirmed by the fact that the failure of board samples during the IB test occurred precisely in the places of EPS granules concentration; veneer delamination was not observed. Some authors [3] also found that IB values of lightweight boards with a density of 550 kg/m^3^ meet the minimum requirements of the EN 312 standard for conventional particleboards type P2 (0.35 N/mm^2^). Referring to Figure 9 and Figure 10, IB strength is closely linked to the density in the core layer, while bending properties (MOR and MOE) are closer associated with the density of veneer in the surface layer of the board. The results of this study are in good agreement with the results of Chow et al. [45] who showed that wood veneer coating did not affect internal bonding.

TS and WA values of the samples are illustrated in Figure 11. It was found that the veneered boards swell less than non-veneered boards. The TS values after 24 h in water of non-veneered boards were higher by 24.4%, 26.4% and 10.0% than those values of veneered boards for densities of 350, 450 and 550 kg/m^3^, respectively (Figure 11a). This is explained by the inclusion of veneer in the surface layers of veneered boards. In this case, the layer of veneer prevents the penetration of water into the porous structure of lightweight boards. It is also observed a higher TS by an increased density of boards that can be explained by the increased proportion of swellable wood material. Kawai et al. [30] also observed a little less thickness swelling in the veneer-overlaid particleboards of densities 300–400 kg/m^3^. On the contrary, some other authors [3] observed that boards with lower density have slightly higher TS values due to the higher number of voids in such boards. This is also related to the long-term effect of water, which destroys the adhesive bonds in the boards, weakens the internal structure, increases the number of voids and water permeability in the middle of the boards and, as a result, increases the swelling. With increasing the density of boards, the swelling increases due to the increased amount of wood component per unit of board volume. In this instance, the protective function of the outer veneer layer in the board gradually decreases. Other researchers also confirm this. In particular, Dziurka et al. [11] found that increased bonding strength of the boards using rapeseed straw and EPS was accompanied by a decrease in WA and TS values. It is the authors’ opinion that the improved hydrophobic properties of the boards were achieved by using isocyanate adhesive and EPS. The TS values were increased with increasing the densities from 350 to 550 kg/m^3^ for both non-veneered and veneered boards (Figure 11a). However, the values of TS for densities 450 and 550 kg/m^3^ differ insignificantly (*p* > 0.05) based on Duncan’s test.

The WA values were decreased by increasing the densities from 350 to 550 kg/m^3^ for both non-veneered and veneered boards (Figure 11b). The less voids exist in the boards with higher density, so the absorption of water is less. The average values of WA for non-veneered boards were higher by 24.7%, 14.9% and 10.9% than those average values of veneered boards for densities of 350, 450 and 550 kg/m^3^, respectively. The lower values of WA for veneered boards may be explained by the presence both of hydrophobic EPS and veneer sheets in these boards. Previous authors had also shown that properties (except for IB strength) of boards can be improved with different coatings including wood veneer [27,29,45]. It is also worth emphasizing the ecological aspect of veneering. It is known that veneering is effective for reducing the emissions of volatile organic compounds and formaldehyde of the particleboards. The formaldehyde emissions in veneered particleboard were less than that in non-veneered board [28].

## 4. Conclusions

The findings of this study suggest that improvement in the properties of lightweight particleboards manufactured with EPS granules in the core layer can be achieved by the incorporation of wood veneers in their surface layers. Veneering of lightweight boards significantly improves MOR and MOE, but does not affect IB. Veneered boards swell less and absorb less water than non-veneered ones.

The density and content of EPS granules (with the exception of TS and IB) significantly affect the properties of lightweight veneered particleboards. In addition, the effect of density was stronger than the effect of EPS content. As the board density decreases, MOR and MOE decrease too. Making boards with a density of 350 kg/m^3^ reduces MOR by 2.64 times, MOE by 1.64 times, IB by 2.53 times, TS by 1.35 times and WA increases by 1.68 times compared to a density of 550 kg/m^3^. With a lower density, but a higher content of EPS, the lightweight veneered boards are more elastic. A change in EPS content between 4% and 10% does not affect TS and IB. As the density and EPS content increase, the WA of lightweight boards decreases. An increase in the pressing temperature from 200 to 220 °C has a negative effect on MOR, but a positive effect on IB.

A relevant comparison of bending and bonding properties values with those in technical standards showed that the minimum requirements, according to the European standards EN 312 (P2 and P3 boards), for conventional particleboards were fulfilled. Moreover, the MOR and MOE of veneered boards throughout the density range (350–550 kg/m^3^) meet the requirements of the standard CEN/TS 16368 for lightweight particleboards types LP1 and LP2. The IB strength of veneered boards with densities of 450 and 550 kg/m^3^, as well as non-veneered boards with a density of 550 kg/m^3^ meet the requirements of CEN/TS 16368 standard for boards type LP1 only.

## Figures and Tables

**Figure 1 materials-15-06474-f001:**
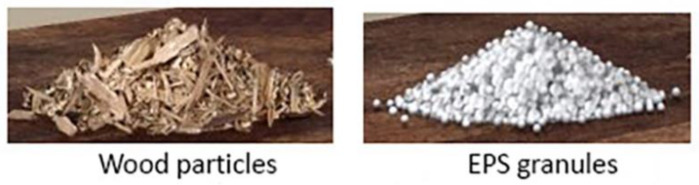
Wood particles and EPS granules used for manufacture of lightweight particleboards.

**Figure 2 materials-15-06474-f002:**
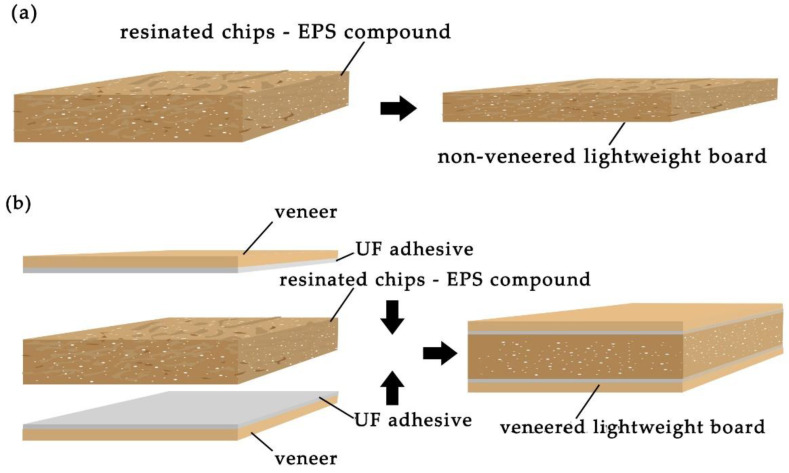
Structure of non-veneered (**a**) and veneered (**b**) lightweight particleboards.

**Figure 3 materials-15-06474-f003:**
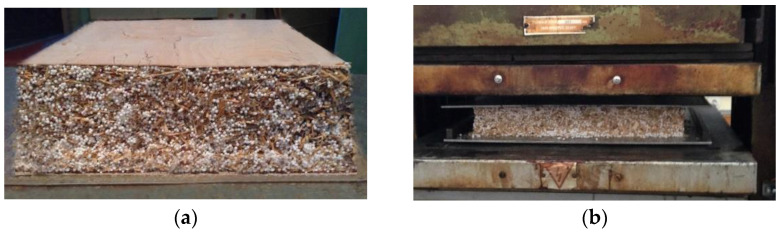
Pre-pressed mat (**a**) and the board pressing in the hot press (**b**).

**Figure 4 materials-15-06474-f004:**
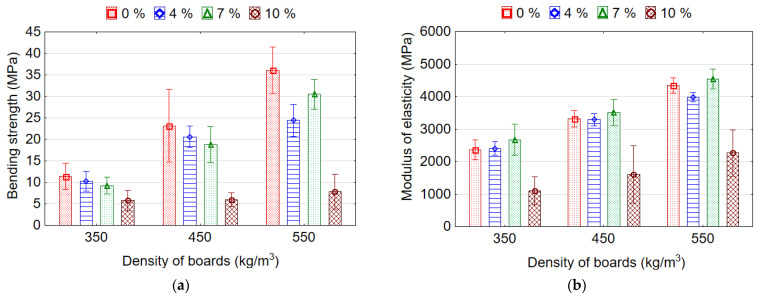
Bending strength (**a**) and modulus of elasticity (**b**) of lightweight veneered particleboards.

**Figure 5 materials-15-06474-f005:**
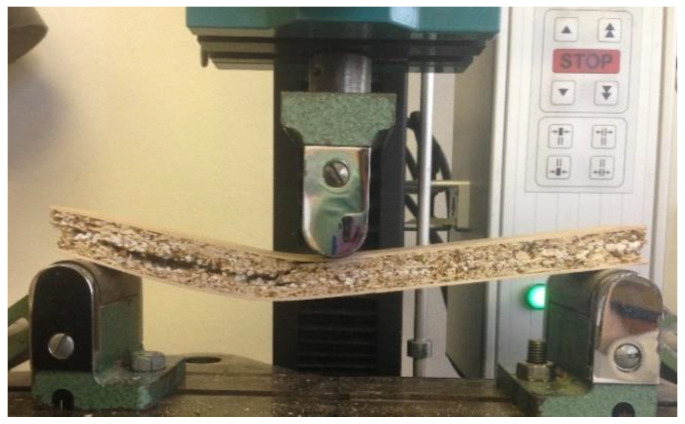
Delamination of the samples under bending strength test.

**Figure 6 materials-15-06474-f006:**
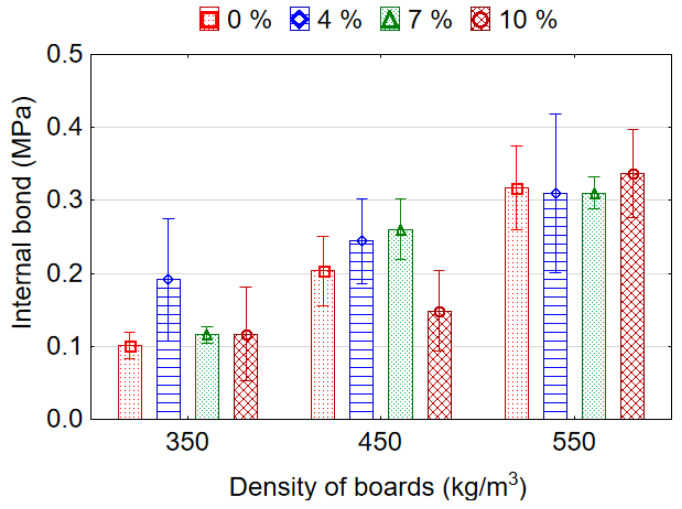
The internal bond strength of the lightweight particleboards.

**Figure 7 materials-15-06474-f007:**
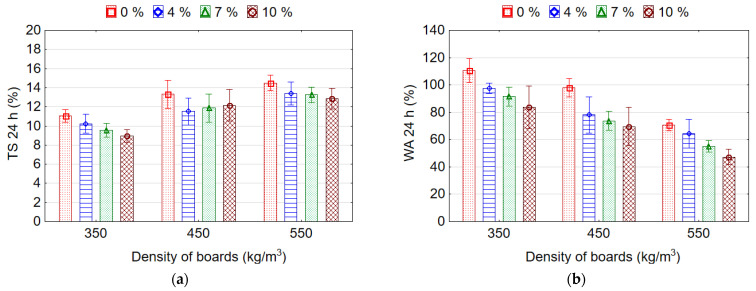
Thickness swelling (**a**) and water absorption (**b**) of the lightweight particleboards.

**Figure 8 materials-15-06474-f008:**
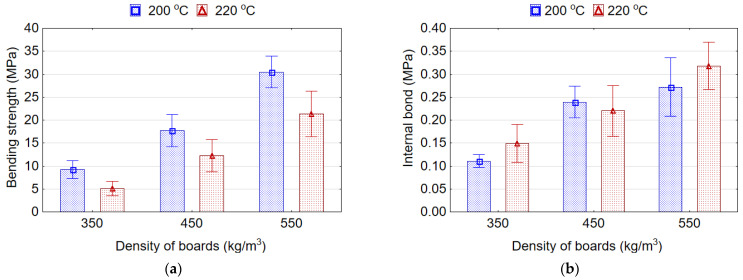
The bending strength (**a**) and internal bond strength (**b**) of the lightweight particleboards.

**Figure 9 materials-15-06474-f009:**
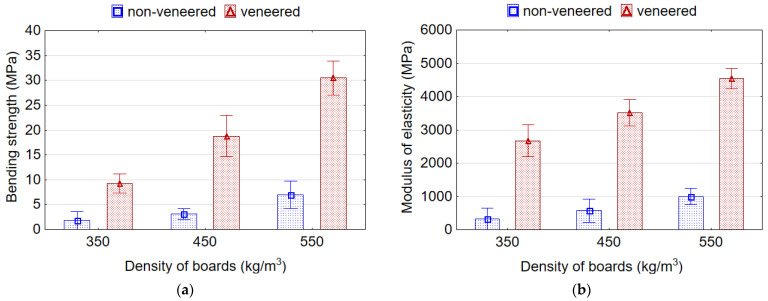
Bending strength (**a**) and modulus of elasticity (**b**) of non-veneered and veneered lightweight particleboards.

**Figure 10 materials-15-06474-f010:**
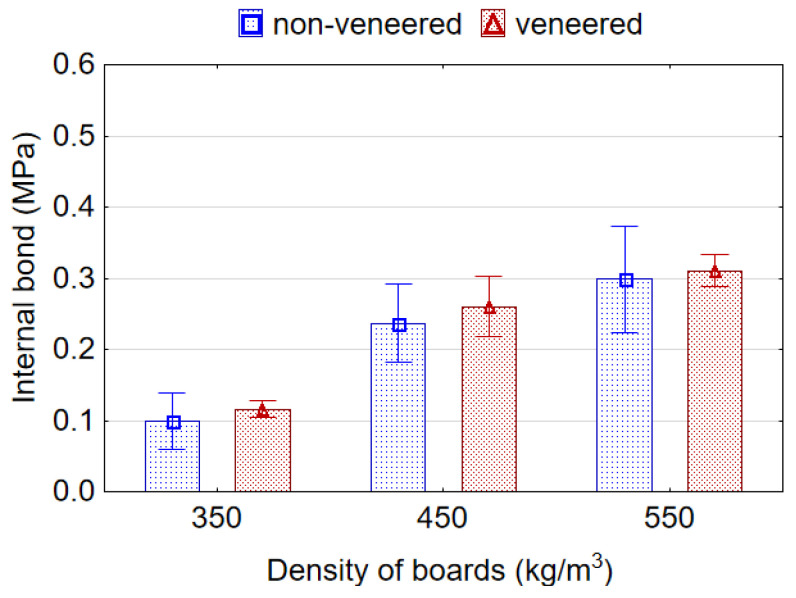
Internal bond strength of non-veneered and veneered lightweight particleboards.

**Figure 11 materials-15-06474-f011:**
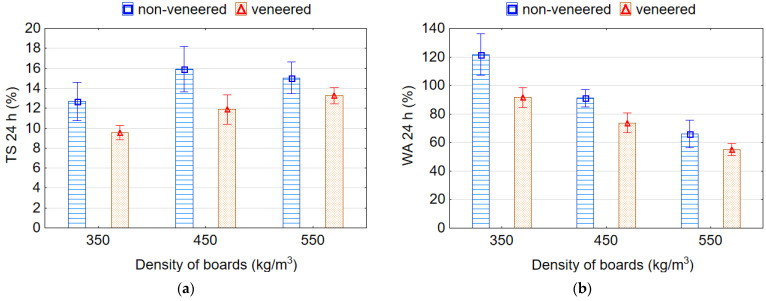
Thickness swelling (**a**) and water absorption (**b**) of non-veneered and veneered lightweight particleboards.

**Table 1 materials-15-06474-t001:** List of boards manufactured and symbols used.

Code	Density of Boards (kg/m^3^)	EPS Content (%)
1	350	0
2	450	0
3	550	0
4	350	4
5	450	4
6	550	4
7	350	7
8	450	7
9	550	7
10	350	10
11	450	10
12	550	10

**Table 2 materials-15-06474-t002:** Time needed for the core layer of mat to reach 100 °C.

Content of EPS (%)	Time to Reach 100 °C (s)		
	Density of Boards (kg/m^3^)		
	350	450	550
0	100	93	90
4	93	78	63
7	88	83	60
10	98	93	70

**Table 3 materials-15-06474-t003:** Some properties of lightweight veneered particleboards compared to the CEN/TS 16368 standard.

Property	Density of Boards (kg/m^3^)									Requirements According to CEN/TS 16368 for Board Types	
	350			450			550				
	Content of EPS (%)										
	4	7	10	4	7	10	4	7	10	LP1	LP2
Bending strength (MPa)	10.3	9.2	5.8	20.6	18.8	6.0	24.4	30.5	7.9	≥3.5	≥7.0
Modulus of elasticity (MPa)	2405	2670	1097	3289	3519	1606	3983	4540	2262	≥500	≥950
Internal bond (MPa)	0.19	0.12	0.12	0.24	0.26	0.15	0.31	0.31	0.34	≥0.24	≥0.35
Thickness swelling 24 h (%)	10.2	9.6	8.9	11.5	11.9	12.2	13.4	13.3	12.9	is not regulated	

## Data Availability

Not applicable.

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
