# Peer review of "Selected Properties of Veneered Lightweight Particleboards with Expanded Polystyrene"

_materials, 2022, doi:10.3390/ma15186474_

Round 1

Reviewer 1 Report

The authors presented results on some selected properties of birch veneered particleboards containing expanded polystyrene (EPS) in the core layer. The studies were focused mostly on the performance of the lightweight particleboards. Explanation of the observed changes of parameters was presented in the light of previous studies. Yet, sometimes the literature data may be inconsistent with the observed results. Unfortunately, no microscopic insight was made by the authors themselves on the possible mechanisms that could operate in the studied materials. I think such discussion is very important as long as a manuscript is designed as a scientific paper and not only a technical report.

For example, changes in the water adsorption and permeability were discussed for changes in the density and interactions between wood particles. Yet no experimental data were provided to analyze changes in the interactions between the boards’ material and EPS particles. Those interactions are very important for the water permeability (percolation effect) as well as mechanical properties. Please comment.

Please comment also on changes of flammability of the boards depending on the content of EPS. The softening temperature of EPS should be given for the discussion of the effect of pressing temperature.

The author used two ways of presenting quantitative the changes in parameters (e.g. “With decreasing density of samples from 550 to 350 kg/m3 and from 550 to 450 kg/m3 the average values of MOR decreases by 2.64 (164.5%) and 1.54 (54.4%) times, respectively.” – lines 220-221). I think that it is not necessary and only % should be used.

Reviewer 2 Report

This article showed particleboards with different loading of expanded polystyrene laminated with veneer sheet . Novelty factor was somewhat lacked in this work. Overall, I recommend the publication of this manuscript with comments below.

No critical discussion on the results is presented. The author merely reports on the increasing and decreasing of the results. If possible, please include supporting evidence; SEM, FTIR, XRD.

How and why the expanded polystyrene improve the performance of particleboard are not explained and mentioned. "During the pressing of boards, the EPS granules contribute to the compaction of particles and the increase of inter-particles adhesive contacts." This was written by the author without any evidence to support the statement.

Why these two pressing temperatures 200 °C to 220 °C were selected.

This is written by the author to support the need of this study "Nowadays, there is very little research on the veneering of lightweight boards using EPS granules." But only one veneer thickness was selected; 1.5mm. Please explain why.

Several bad expressions are found (as attached).

Reviewer 3 Report

The work presents relevance and importance. The introduction presents the main problem and the research is well justified. The methodologies are correct. The results are clearly presented and well discussed. it is recommended in the conclusion
should be shorter, a discussion of the results is observed in the conclusions. Authors must conclude directly, according to the proposed objectives. Approval of the work is recommended.

Round 2

Reviewer 1 Report

The manuscript may be accepted for publication.